# Dental Pulp Stem Cells Modulate Inflammasome Pathway and Collagen Deposition of Dermal Fibroblasts

**DOI:** 10.3390/cells13100836

**Published:** 2024-05-14

**Authors:** Giada Zanini, Giulia Bertani, Rosanna Di Tinco, Alessandra Pisciotta, Laura Bertoni, Valentina Selleri, Luigi Generali, Alessandra Marconi, Anna Vittoria Mattioli, Marcello Pinti, Gianluca Carnevale, Milena Nasi

**Affiliations:** 1Department of Life Sciences, University of Modena and Reggio Emilia, 41125 Modena, Italy; giada.zanini@unimore.it (G.Z.);; 2Department of Surgical, Medical, Dental and Morphological Sciences, University of Modena and Reggio Emilia, 41125 Modena, Italy; giulia.bertani@unimore.it (G.B.); rosanna.ditinco@unimore.it (R.D.T.); alessandra.pisciotta@unimore.it (A.P.); laura.bertoni@unimore.it (L.B.); luigi.generali@unimore.it (L.G.); alessandra.marconi@unimore.it (A.M.); gianluca.carnevale@unimore.it (G.C.); milena.nasi@unimore.it (M.N.); 3National Institute for Cardiovascular Research—INRC, 40126 Bologna, Italy; annavittoria.mattioli@unibo.it; 4Department of Medical and Surgical Sciences for Children and Adults, University of Modena and Reggio Emilia, 41125 Modena, Italy

**Keywords:** fibrosis, fibroblasts, dental pulp stem cells, inflammasome, inflammation, fibrosis

## Abstract

Fibrosis is a pathological condition consisting of a delayed deposition and remodeling of the extracellular matrix (ECM) by fibroblasts. This deregulation is mostly triggered by a chronic stimulus mediated by pro-inflammatory cytokines, such as TNF-α and IL-1, which activate fibroblasts. Due to their anti-inflammatory and immunosuppressive potential, dental pulp stem cells (DPSCs) could affect fibrotic processes. This study aims to clarify if DPSCs can affect fibroblast activation and modulate collagen deposition. We set up a transwell co-culture system, where DPSCs were seeded above the monolayer of fibroblasts and stimulated with LPS or a combination of TNF-α and IL-1β and quantified a set of genes involved in inflammasome activation or ECM deposition. Cytokines-stimulated co-cultured fibroblasts, compared to unstimulated ones, showed a significant increase in the expression of IL-1β, IL-6, NAIP, AIM2, CASP1, FN1, and TGF-β genes. At the protein level, IL-1β and IL-6 release as well as FN1 were increased in stimulated, co-cultured fibroblasts. Moreover, we found a significant increase of MMP-9 production, suggesting a role of DPSCs in ECM remodeling. Our data seem to suggest a crosstalk between cultured fibroblasts and DPSCs, which seems to modulate genes involved in inflammasome activation, ECM deposition, wound healing, and fibrosis.

## 1. Introduction

Fibrosis is a pathological condition that can affect almost every tissue of the body, consisting of a delayed deposition and remodeling of the extracellular matrix (ECM) by fibroblasts. It is related to persistent infections, genetic disorders, repeated exposures to chemical substances and toxins, metabolic disorders (hypertension, hypercholesterolemia, diabetes, obesity), and minor human-leukocyte antigen mismatches in transplants [1,2,3,4]. Fibrosis, which often arises from a damage-repairing process, consists of two phases: a regenerative phase characterized by the replacement of injured cells with new ones of the same cytotype, and a fibroplasia phase characterized by the replacement of the normal parenchymal tissue with connective tissue [5,6]. Fibrosis is mainly triggered by a chronic stimulus mediated by pro-inflammatory cytokines. Tumor necrosis factor-alpha (TNF-α) and interleukin-1β (IL-1β) can act synergistically, initiating the inflammatory cascade. These cytokines, released by activated macrophages in addition to transforming growth factor (TGF)-β, have been shown to activate fibroblasts, with an overproduction of ECM proteins, endothelial cells, for neo-angiogenesis processes, keratinocytes, and macrophages [7].

Fibroblasts are mesenchymal-derived cells present in the connective tissue, whose main role is to preserve tissue homeostasis by regulating the renewal of ECM [8,9]. They are also involved in key processes such as tissue remodeling, wound healing, inflammation, angiogenesis, and cancer [10,11,12]. During the wound healing process, fibroblasts migrate to the wound site in response to growth factors secreted by macrophages, such as TGF-β and produce newly formed ECM. Moreover, fibroblasts undergo a phenotypic switch in myofibroblasts, with contractile properties given by α-Smooth-Muscle-Actin (α-SMA) [13,14]. This process of fibroblast differentiation is induced by their exposure to TGF-β1 and TGF-β2 [1,3]. This transition is crucial in the wound healing process since myofibroblasts are responsible for tissue contraction and production of ECM, such as collagen type 1, that permit the formation of the scar [15,16,17]. Myofibroblast ECM deposition is regulated by a balance between matrix metalloproteinases (MMPs), a family of enzymes able to cleave basal membranes, ECM components [18,19], processing bioactive mediators such as cytokines, chemokines, growth factors, and cell-surface receptors [20,21], together with tissue inhibitors of matrix metalloproteinases (TIMPs), a family of four MMPs inhibitors [22]. Thus, an imbalance between MMPs and TIMPs, with the consequent excess of collagen deposition, is one of the most contributing factors to the onset of fibrosis, as well as the persistent myofibroblast activation and chronic inflammatory response [5].

The main aspects of inflammation and tissue repair, such as fibrogenesis, are also regulated by inflammasomes [23,24,25]. Inflammasomes are a group of multimeric protein complexes, composed of a sensor, an adaptor, and a caspase recruitment domain, capable of recognizing different sets of inflammation-inducing stimuli and controlling activation of the proteolytic enzyme caspase-1 (CASP-1), which regulates maturation of the pro-inflammatory cytokines (IL-1β) and IL-18, leading to pyroptosis [23,26,27,28]. They mediate the recruitment and activation of inflammatory cells to the site of danger through the release of pro-inflammatory factors. Furthermore, inflammasomes are able to recognize exogenous and endogenous alarm signals [23,29,30]. The most studied inflammasomes are NOD-, LRR-, and pyrin domain-containing protein 3 (NLRP3) and absent-in-melanoma 2 (AIM2). NLRP3 inflammasome senses several stimuli including pore-forming toxins, extracellular ATP, RNA-DNA hybrids, and infective pathogens such as viruses, bacteria, protozoa, and fungi. NLRP3 assembly is finely regulated and needs two activation signals, usually the engagement of pattern recognition receptors (PRRs), such as Toll-like receptor 4 (TLR4) or nucleotide-binding oligomerization domain-containing protein 2 (NOD2) [23,31]. Moreover, non-canonical pathways can activate NLRP3, such as potassium efflux mediated by pore-forming proteins and ion channels [32,33] and mitochondrial damage resulting in the release of mitochondrial reactive oxygen species (mtROS) and mitochondrial DNA (mtDNA) [33,34].

AIM2 inflammasome is a cytoplasmic sensor that binds cytosolic double-strand DNA (dsDNA) [23]. AIM2 recognizes dsDNA from naïve and recombinant strains (used in vaccine formulations) [35,36,37,38]. AIM2 activation also plays an important role in the regulation of cancer cell proliferation and tumorigenesis [39].

Due to their anti-inflammatory and immunosuppressive potential, mesenchymal stem cells (MSCs) have become a promising tool for developing therapeutic strategies to many chronic inflammatory diseases. Among different tissue sources of MSCs, human dental pulp is a reservoir of ectomesenchymal stem cells, i.e., human dental pulp stem cells (hDPSCs) first characterized in 2002 by Gronthos et al. [40]. In light of their embryonic origin from neural crest, hDPSCs own peculiar features, such as high proliferative rate, a wide differentiation potential in vitro and in vivo [41,42,43,44], and low immunogenicity and immunomodulatory properties of note, the latter ones being exerted on immune cells through the activation of different mechanisms, such as Fas/Fas ligand (FasL) and programmed death-protein 1 (PD1)/PD1 ligand 1 (PD-L1) pathways, and the release of soluble factors [45,46,47,48]. The immunomodulatory role of MSCs, in particular that of DPSCs in fibrosis, is still far from being understood [49].

In the present study, we analyzed the potential of hDPSCs in modulating the activation of the inflammasome and the fibrotic activity of fibroblasts in the context of a pro-inflammatory environment, mimicked by the stimulation with TNF-a + IL-1β, or by the presence of lipopolysaccharide (LPS), which induces the switch of normal skin fibroblasts into hypertrophic scar tissue fibroblasts and increases fibroblast migration.

## 2. Materials and Methods

### 2.1. Isolation and Characterization of Human Fibroblasts and Dental Pulp Stem Cells

Primary human fibroblasts were isolated from the foreskin of healthy male donors, from waste materials obtained from surgery (n = 3). All samples were collected with a written informed consent of patients, according to the Declaration of Helsinki after approval of the Area Vasta Emilia Nord Ethical Committee (ref. number 184/10). Briefly, skin biopsies were surgically removed, and fibroblasts were obtained by explant culture and grown in Dulbecco’s modified Eagle’s medium (DMEM) containing 5% fetal bovine serum (FBS), 2 mM L-glutamine, and 1% of penicillin/streptomycin (all from Sigma Aldrich, St. Louis, MO, USA). Human fibroblasts were frozen at a low number of passages (1–2) to preserve their longevity.

DPSCs were isolated from the dental pulp of the third molars of three male patients (18–25 years) as previously described [43]. Briefly, the enzymatic digestion of the harvested dental pulp was carried out by using 3 mg/mL type I collagenase and 4 mg/mL dispase (Sigma-Aldrich Saint Louis, MO, USA) in alpha minimum essential medium (α-MEM). Cells were filtered by using 100 µm Falcon Cell Strainers, then the suspension was plated and expanded at 37 °C and 5% CO_2_ in α-MEM with 10% of FBS, 2 mM L-glutamine, and 1% of penicillin/streptomycin (Sigma-Aldrich Saint Louis, MO, USA). After reaching 80% of confluency, immuno-selection against STRO-1 and c-Kit was performed on the isolated DPSCs by using the magnetic activated cell sorting (MACS)^®^ separation kit (Miltenyi Biotec, Gladbach, Germany), according to the manufacturer’s instructions. Approximately 5 × 10^6^ DPSCs underwent double separation: cells were first selected by using the primary antibody mouse IgM anti-STRO-1 (Santa Cruz Biotechnology Santa Cruz, Dallas, TX, USA) detected by the following magnetically labeled secondary antibodies anti-mouse IgM (Miltenyi Biotec, Bergisch Gladbach, Germany). Then, DPSCs were expanded until 80% of confluency, and newly sorted by using rabbit IgG anti-c-Kit (Santa Cruz Biotechnology Santa Cruz, Dallas, TX, USA) and magnetically labeled anti-rabbit IgG (Miltenyi Biotec, Bergisch Gladbach, Germany). These immune-selected hDPSCs at passage 1 underwent an immunophenotypical characterization by assaying through flow cytometry their expression of mesenchymal stem cells (MSCs) markers, i.e., CD73, CD90, CD105, CD34, CD45, and at the same time the lack of expression of HLA-DR [43]. STRO-1+/c-Kit+ DPSCs were used for all the experiments.

The study was carried out according to the recommendations of Comitato Etico Provinciale-Azienda Ospedaliero-Universitaria di Modena (Modena, Italy), which provided the approval of the protocol (ref. number 3299/CE; 5 September 2017).

### 2.2. Cell Culture

Fibroblasts and DPSCs were seeded on six-well plates at a density of 2.5 × 10^5^ and 2 × 10^5^, respectively, in α-MEM with 10% FBS, 2 mM L-glutamine, and 1% of penicillin/streptomycin. In parallel, indirect co-cultures were established between fibroblasts and hDPSCs as follows. Fibroblasts and hDPSCs from donors were randomly matched. Fibroblasts were first seeded on the bottom of 12-well culture plate at a cell density of 2.5 × 10^5^ cells/cm^2^; then, 24 h later, hDPSCs were seeded at a cell density of 2 × 10^5^ cells/cm^2^ on transwell inserts—0.4 μm polycarbonate membranes (Corning, New York, NY, USA)—and placed above the previously seeded fibroblasts. Then, cells cultured alone and in indirect transwell co-culture, were stimulated with either 1 µg/mL lipopolysaccharide (LPS; Sigma-Aldrich Saint Louis, MO, USA) or a combination of 10 ng/mL TNF-α and 10 ng/mL IL-1β (both from R&D Systems, Minneapolis, MN, USA). Cells were collected after 4 h for gene expression analysis and after 24 h for protein expression analysis. Fibroblasts and hDPSCs, either cultured alone or in indirect transwell co-culture without exposure to any stimuli, were used as controls. When indicated, fibroblasts were also treated with DPSC supernatant (either treated or not with 10 ng/mL TNF-α and 10 ng/mL IL-1β) in the presence of dexamethasone (DEX; Sigma-Aldrich) 1 µM for 4 or 24 h. All the experiments were performed in triplicate.

### 2.3. RNA Extraction, Reverse Transcription of mRNA, and Real-Time PCR

Total RNA was extracted from cells by QuickRNA miniPrep kit (Zymo Research, Irvine, CA, USA) following the manufacturer’s instructions. The concentration of RNA was quantified using the NanoDrop ND-1000 (Thermo Fisher Scientific, Waltham, MA, USA). Then, 1 µg of RNA was reverse transcribed using an iScript cDNA synthesis kit from (Bio-Rad, Hercules, CA, USA). Gene expression analysis was performed by quantitative real-time PCR by using a CFX96 Touch Detection System (Bio-Rad, Hercules, CA, USA). Fifteen genes involved in the inflammasome activation and in the deposition of fibrotic scar tissue were detected using pre-validated Prime PCR Assay (Bio-Rad, Hercules, CA, USA): *RPS18* (qHsaCED0037454) was the reference gene, *AIM2* (qHsaCID0018402), *IL1B* (qHsaCID0022272), *IL18* (qHsaCID0006163), *NAIP* (qHsaCID0038447), *NLRP3* (qHsaCID0036694), *PYCARD* (qHsaCED0042977), *COL1A1* (qHsaCED0043248), *COL3A1* (qHsaCED0046560), *TGFB1* (qHsaCID0017026), *FN1* (qHsaCED0043611), *CASP1* (qHsaCED0038605), *TLR4* (qHsaCED0037607), *IL6* (qHsaCID0020314), *CD274* (qHsaCID0036468) and *PDCD1* (qHsaCID0014211). Gene relative expression was calculated through the ΔΔ-cycle method.

### 2.4. Western Blot

Whole-cell extracts were prepared as previously described [46,50], and protein concentration was determined by Bradford Protein Assay (Bio-Rad, Hercules, CA, USA). Thirty μg of protein was extracted per sample, run on SDS-polyacrylamide gel electrophoresis, and then transferred to nitrocellulose membranes. The following antibodies were used: mouse anti-α smooth muscle actin (αSMA; Cat. #MA1-06110, Invitrogen, Waltham, MA, USA), mouse anti-Fibronectin (FN1; Cat. #MA5-14737, Invitrogen), mouse anti-MMP9 (Cat. #SMC-396, StressMarq Biosciences, Victoria, BC, Canada), rabbit anti-PD-L1 (Cat. #NBP2-15791; Novus Biologicals, Centennial, CO, USA), rabbit anti-PDGFRβ (Cat. #3169S; Cell Signaling Technology, Danvers, MA, USA). Protein detection was revealed by using Clarity™ Western ECL Substrate (Bio-Rad, Hercules, CA, USA). Rabbit anti-GAPDH and anti-actin antibodies (Cat. #SPC-689, #SPC-687D, StressMarq Biosciences, Victoria, BC, Canada) were used as controls of protein loading. Densitometry analysis was performed through the open-access software platform FIJI (ImageJ version 1.53q, National Institute of Health, MD, USA). Data were normalized to values of background and of control GAPDH band for FN1, MMP-9, and α-SMA. Data were analyzed by ANOVA followed by the Newman–Keuls post hoc test and expressed as mean ± standard deviation (SD) of three independent experiments.

### 2.5. Confocal Microscopy

For immunofluorescence analyses, cells cultured alone or in indirect co-culture were seeded on glass coverslips and cultured for 24 h. Then, cells were fixed with 4% paraformaldehyde (PFA) in pH 7.4 PBS and underwent immunolabeling with the following primary antibodies: mouse anti-α smooth muscle actin (αSMA; Cat. #MA1-06110, Invitrogen, Waltham, MA, USA), mouse anti-Fibronectin (FN1; Cat. #MA5-14737, Invitrogen, Waltham, MA, USA), mouse anti-MMP9 (Cat. #SMC-396, StressMarq Biosciences, Victoria, BC, Canada), rabbit anti-COL1A1 (Cat. #72026S, Cell Signaling Technology, Trask Lane Danvers, MA, USA), rabbit anti-SOX10 (Cat. #ab155279; Abcam, Cambridge, UK), mouse anti-nestin (Cat. #MAB5326; Millipore, Burlington, MA, USA), rabbit anti-PDGFRb (Cat. #3169S; Cell Signaling Technology; Danvers, MA, USA), mouse anti NF-kB p65 (Cat. #33-900; Invitrogen), subsequently revealed by Alexa Fluor-conjugated secondary antibodies (goat anti-mouse Alexa488, goat anti-rabbit Alexa 546, goat anti-mouse Alexa546, all diluted 1:200; Thermo Fisher, Waltham, MA, USA). For cell morphology analyses, hDPSCs were labeled with Phalloidin iFluor 555 reagent (Cat. #ab176756; Abcam, Cambridge, UK).

After counterstaining cell nuclei with 1 μg/mL 4′,6-diamidino-2-phenylindole dihydrochloride (DAPI), samples were mounted with an anti-fading medium (FluoroMount, Sigma-Aldrich). Images were then acquired on a confocal laser scanning microscope Nikon A1 (Nikon) and image analysis was performed as previously described [41]. Quantification of NF-kB nuclear translocation in fibroblasts was performed on 2D confocal microscopy images using Fiji (ImageJ). Briefly, black and white images for both NF-kB and DAPI signals were cleared for the background. Then, DAPI and NF-kB signal areas were manually selected, and the threshold was adjusted. Subsequently, nuclear and cytosolic NF-kB signals were automatically calculated. The ratio between the signal intensity and the area considered was calculated.

### 2.6. Quantification of Cytokines in Cell Supernatants

Quantification of pro-inflammatory cytokines IL-1β, IL-6, and TNF-α in fibroblast supernatant was performed by using ELLA Multianalyte assay (Biotechne, San Jose, CA, USA) following the provided instructions, as described [51]. Data are expressed as pg/mL.

### 2.7. Statistical Analysis

Statistical analyses were performed using Prism 9.2.1 (GraphPad Software, La Jolla, CA, USA). One-way ANOVA was followed by the Bonferroni and Newman–Keuls post hoc tests to compare quantitative variables. A *p*-value < 0.05 was considered significant. All measurement data are presented as mean ± SD of three independent experiments, unless otherwise specified.

## 3. Results

We set up a transwell co-culture system to mimic an in vivo pro-inflammatory environment, where DPSCs were seeded above a fibroblast monolayer on the well bottom and stimulated with LPS or a combination of the pro-inflammatory cytokines TNF-α and IL-1β. The relative expression of genes involved in the inflammasome activation, or the deposition of fibrotic scar tissue, was quantified by quantitative real-time PCR, after four hours of stimulation.

### 3.1. DPSCs Modulated the Expression of Pro-Inflammatory Cytokines in Fibroblasts

We first analyzed the expression of pro-inflammatory cytokines in fibroblasts treated with LPS (Figure 1). As expected, treatment of fibroblasts determined a huge increase in the transcription of *IL1B*, whose levels were 28.3 fold higher than unstimulated cells (Ctrl group) (Figure 1A). When co-cultured with DPSC, levels of IL-1β after treatment with LPS increased up to 102-fold if compared to untreated cells, suggesting that LPS, as expected, enhanced the expression of IL-1β when cells are in co-culture (Figure 1A). Concerning *IL6* expression, a statistically significant upregulation was detected in fibroblasts stimulated in co-culture with DPSCs (ccS) for 4 h when compared to fibroblasts stimulated alone (S) (Figure 1A), suggesting a direct contribution of DPSCs in *IL6* increase when exposed to LPS. Finally, LPS did not seem to significantly modify the expression of *IL18* in any of the tested conditions. At least in part, these effects could be attributed to the modulation of *TLR4* expression, which is increased by stimulation with LPS and TNF-α + IL-1β (Appendix A).

When fibroblasts were stimulated with TNF-α and IL-1β, the expression of *IL1B* and *IL6* was sharply increased in stimulated fibroblasts in co-culture with hDPSCs, concerning the stimulated fibroblasts cultured alone (Figure 1B). Interestingly, when evaluating IL-18 mRNA expression levels, a statistically significant increase was detected in stimulated fibroblasts when compared to the control group (NS) (Figure 1B). After co-culture with DPSCs in the presence of stimuli, IL-18 showed a statistically significant downregulation (Figure 1B). These data suggest that hDPSCs can exert an immunomodulatory effect on pro-inflammatory *IL18* expression in fibroblasts still under exposure to TNF-α and IL-1β.

### 3.2. Pro-Inflammatory Stimuli Upregulated AIM2, but Not NLRP3, in Fibroblasts

We next wondered whether treatment with LPS or TNF-α and IL-1β could also modify the expression of inflammasomes and determine a higher release of pro-inflammatory cytokines. Thus, we evaluated the expression of *AIM2*, *PYD*, and CARD domain containing (*PYCARD*), NLR apoptosis inhibitory protein (*NAIP*), *NLRP3*, and *CASP1* genes.

As shown in Figure 2, treatment with LPS caused a huge increase in the expression of *AIM2* and *CASP1* in fibroblasts alone (Figure 2A); co-culture with DPSCs increased this effect of three-fold, when compared to both fibroblasts non-stimulated in co-culture with DPSCs (ccNS) and to S (Figure 2A). Conversely, *NLRP3* appeared barely expressed in fibroblasts, and LPS did not affect its expression (Appendix A, left panel).

Stimulation with TNF-α and IL-1β provoked an increase in the expression of *AIM2* in fibroblasts, which is enhanced by co-culture with DPSCs (Figure 2B). Concerning CASP1, co-culture with DPSCs caused an increase in its expression after stimulation with TNF-α and IL-1β if compared to fibroblasts alone (Figure 2). Also, in this case, NLRP3 was not affected by TNF-α and IL-1β (Appendix A, right panel). No significant effects on the expression of *PYCARD* or *NAIP* could be observed after treatment with LPS or pro-inflammatory cytokines in fibroblasts alone, or co-cultured with DPSCs (Appendix A). We also inspected the mRNA levels of pro-inflammatory cytokines by DPSCs, under the same experimental conditions (Appendix A). The expression of IL-1β and IL-18 was barely detectable, both in basal conditions and after stimulation, in these cells. However, they did express IL-6, whose levels are strongly upregulated by TNF-α and IL-1β. Taken together, these data suggest that stimulation with either LPS or TNF-α and IL-1β induced the activation of the pro-inflammatory conditions in fibroblasts and that co-culture with DPSCs had a promoting effect on the inflammatory milieu.

### 3.3. Release of Inflammatory Cytokines by Fibroblasts after Stimulation with TNF-α and IL-1β

To determine if the upregulation of the genes led to a higher cytokine release by fibroblasts, we evaluated their levels in the supernatant of stimulated cells by ELLA. After 4 h of stimulation with TNF-α + IL-1β, fibroblasts alone showed an increase of TNF-α, IL-1β, and IL-6 expression levels (Figure 3A). Interestingly, a statistically significant decrease of TNF-α was observed in stimulated fibroblasts cultured with DPSCs with respect to stimulated fibroblasts cultured alone (Figure 3A), whereas an up-regulation of IL-6 was detected in the same experimental group (Figure 3A).

Interestingly, the same trend was observed after 24 h of stimulation and co-culture with DPSCs (Figure 3B). These data suggest that DPSCs per se can modulate the release of pro-inflammatory cytokines by fibroblasts. No changes were found in the case of IL-18.

### 3.4. DPSCs Modulated the Expression of Genes Involved in Fibrosis

We finally determined whether the co-culture with DPSCs could also modulate the process of fibrosis. Thus, we evaluated the expression of a series of genes crucial for collagen deposition and fibrosis, namely, TGF-β, Fibronectin 1 (*FN1*), Collagen Type I Alpha 1 Chain (*COL1A1*), and Collagen Type III Alpha 1 Chain (*COL3A1*).

As shown in Figure 4, TGF-β levels in fibroblasts were significantly increased only in cell co-culture with DPSCs, regardless of the stimulation with LPS, suggesting that DPSCs produced a factor that can modulate *TGFB* transcription by fibroblasts. When fibroblasts were stimulated with TNF-α and IL-1β, a slight increase in TGF-β expression was detected after co-culture, with respect to either ccNS and S (Figure 4B). With regard to *FN1* expression in fibroblasts after stimulation with LPS, no statistically significant differences were detected among the experimental groups, while the exposure to TNF-α and IL-1β induced a statistically significant upregulation of *FN1* in fibroblasts co-cultured with DPSCs (Figure 4B). As far as collagen genes are concerned, *COL1A1* showed a trend towards an increase in fibroblasts co-cultured with DPSCs, but not in fibroblasts alone when treated with LPS. A significant increase of *COL3A1* was observed in fibroblasts treated with LPS and co-cultured with DPSCs (Figure 5A).

The stimulation with TNF-α and IL-1β induced a statistically significant up-regulation of COL3A1 in comparison with the unstimulated fibroblasts (Figure 5B). Interestingly, co-culture with DPSCs seemed to reduce the expression of *COL3A1* in cells stimulated with TNF-α and IL-1β (Figure 5B). To confirm that the observed modulation of collagen genes was due to the effect of pro-inflammatory stimuli on DPSCs, we treated fibroblasts with DPSC conditioned medium after stimulation with TNF-α and IL-1β, in the presence or absence of anti-inflammatory drug dexamethasone (DEX; Appendix A). The anti-inflammatory effect of DEX was confirmed by the absence of NF-kB translocation to the nucleus (Appendix A) and the lack of *IL1B* and *IL6* upregulation in fibroblasts cultured in the conditioned medium when DEX was present. In these conditions, changes in COL3A1 and COL1A1 previously observed were abolished (Appendix A).

The effects of DPSC co-culture trigger different responses in fibroblasts stimulated with either LPS or TNF-α and IL-1β, in particular, the exposure to LPS induced an enhanced pro-fibrotic milieu, whereas when TNF-α and IL-1β stimulation was carried out DPSCs seem to modulate the expression of ECM components by fibroblasts.

### 3.5. Western Blot and Immunofluorescence Analysis of Proteins Involved in ECM Deposition and Remodeling

To further investigate how LPS or TNF-α + IL-1β stimulation may affect the fibrotic response of fibroblasts, with and without co-culture with DPSCs, the expression of fibrosis markers α-smooth muscle actin (α-SMA) and fibronectin, as well as of matrix metalloproteinase 9 (MMP-9), associated to ECM remodeling, was determined.

No significant variation was observed after treatment with LPS. When treated with TNF-α + IL-1β, fibroblasts showed a decreasing trend in α-SMA expression together with a statistically significant decrease of fibronectin levels in fibroblasts co-cultured with DPSCs (Figure 6A). Noteworthy, our data also highlighted that MMP-9 was statistically significantly upregulated in stimulated fibroblasts after co-culture with DPSCs (Figure 6A). Data from immunofluorescence analyses further confirmed this evidence; as shown in Figure 6B, fibroblasts displayed a decreased expression of α-SMA and COL1A1 when co-cultured with DPSCs, as well as a reduced arrangement of FN1, in parallel with an increased expression of MMP-9. These findings suggest the contribution of DPSCs in modulating the pro-fibrotic phenotype induced in fibroblasts by stimulation with TNF-α + IL-1β (Figure 6B).

### 3.6. Effects of Inflammatory Stimuli on DPSCs

Based on the data shown above, we further investigated whether the stimulation with LPS and TNF-α/IL-1β affected the biological properties and the phenotype of DPSCs. To this purpose, Western blot analyses were performed to assay the expression of platelet derived growth factor receptor β (PDGFRβ), a typical pericyte-marker, and PD-L1, a key immunomodulatory molecule. Our data showed that neither LPS nor TNF-α/IL-1β stimulation induced any significant difference in the expression of PDGFRβ, confirming that the pericyte-like nature of DPSCs was not altered in any experimental group (Appendix A). This evidence was further confirmed by immunofluorescence analysis shown in Appendix A. Interestingly, when TNF-α/IL-1β stimulation was carried out on DPSCs, cultured alone and after indirect co-culture with fibroblasts, PD-L1 expression was increased in a statistically significant manner, when compared to the unstimulated counterpart groups, i.e., DPSCs alone and DPSCs after co-culture (Appendix A). Furthermore, none of the administered stimuli altered the expression of the typical neural crest-related markers, nestin and SRY-box transcription factor 10 (SOX10), by DPSCs nor did they alter stem cell morphology, as highlighted through confocal immunofluorescence analyses (Appendix A).

## 4. Discussion

Inflammation is the first innate response to tissue damage. Although inflammation is crucial in wound healing, a persistent inflammatory response can promote fibrosis. Fibroblasts, through their phenotypic switch, are responsible for ECM deposition and remodeling. This activity is finely regulated by interleukins and growth factors secreted mostly by macrophages in the inflammatory milieu. Recently, it was observed that not only macrophages play a role in fibroblast immunomodulation but also MSCs [49].

Data obtained in our work highlight that similarly to MSC, DPSC can modulate the activity of fibroblasts in an inflammatory microenvironment due to the presence of a combination of proinflammatory cytokines, TNF-α and IL-1β. We also treated fibroblasts and hDPSCs with LPS, which induces the switch of normal skin fibroblasts into hypertrophic scar tissue fibroblasts and increases fibroblast migration in a concentration- and time-dependent manner [52].

The main observation that emerges from the data regarding cytokine expression is that the presence of DPSCs in co-culture induced upregulation of mRNA levels of IL-1β and IL-6 in fibroblasts, and a contemporary downregulation of IL-18. As far as pleiotropic cytokine IL-6 is concerned, increased levels of mRNA are reflected at protein levels, as demonstrated by ELLA assays, further confirming our previous findings [46,47]. As a matter of fact, it is well established that IL-6 plays a central role in acute inflammation and is crucial for wound healing resolution [53]. Interestingly, although IL-1β was increased at mRNA level, hDPSC co-culture was able to modulate its release by fibroblasts reaching a significantly decreased secretion after 24 h. In parallel, TNF-α secretion was reduced as early as after 4 h of co-culture with hDPSCs. When analyzing the expression of the pro-inflammatory cytokine IL-18, we found that DPSC co-culture was able to reduce its mRNA levels, although no significant differences were observed at the protein level. It is well known that IL-18 is constitutively expressed, and its main regulation occurs at the protein level. Although constitutive, *IL18* expression could be induced and requires both NF-kB activation and a type I IFN signaling. This determines a slower kinetics of induction of *IL18* in comparison to *IL1B* or *IL6* and is likely the reason why we did not observe a significant increase of IL18 after 4 h of treatment with LPS, and a slight increase after 4 h of treatment with TNF-α + IL-1β. However, we did not observe a significant increase of mature IL-1β and IL-18 in the supernatant, indicating that the *CASP1* increased transcription determined by DPSC probably does not ultimately impact the release of these cytokines.

We currently do not have direct insights into the mechanisms by which DPSCs modulate fibroblast response in this context. The regulation of IL-1β and IL-18 occurs at protein levels when inflammasomes are formed and CASP1 can cleave pro-IL-1β and pro-IL-18 into their active forms. CASP1 expression is largely constitutive but a loss of balance in CASP1 expression may be relevant in pathological situations [54]. For instance, in dermal fibroblasts from patients with systemic sclerosis, the inhibition of CASP1 reduces the collagen deposition and the IL-18 and IL-1β release [15]. Even if we observed slightly higher levels of CASP1 in the presence of DPSC, this upregulation did not impact the activity of the enzyme, as IL-1β was reduced in cell supernatants by DPSC, and *IL18* was not significantly affected. As far as inflammasomes are concerned, we did not observe significant changes in *NRLP3*, *PYCARD*, and *NAIP*, whose expression was barely detectable in fibroblasts, while stimulation with LPS or proinflammatory cytokines hugely increased AIM2 transcriptional levels. *NLRP3* is usually expressed at low levels in basal conditions. LPS is known to work as a priming signal for NLRP3 inflammasome activation, as it can promote the transcription of *NLRP3* via *TLR4* engagement and activation of NF-kB [55,56]. As we did not observe any increase in *NLRP3* after stimulation with LPS, we can suppose that TLR4 engagement is not effective in activating NLRP3 and is not significantly affected by co-culture with DPSC. This effect cannot be ascribed to TLR4 itself, whose expression in fibroblasts is constant in our model. This was not the case for AIM2, whose levels were hugely induced both by LPS and proinflammatory cytokines. This agrees with the fact that AIM2 levels in humans, differently from mice, are very low at steady state but can be strongly induced by LPS or interferons [57,58]. The upregulation of *AIM2* transcript in inflammatory conditions is likely relevant in human diseases since the *AIM2* expression level is increased in keratinocytes from psoriatic lesions compared to healthy skin [59]. It is also likely that DPSCs secrete factors that act synergistically with LPS or TNF-α +IL-1β, thus increasing the intensity of the signal induced by the latter.

Regarding the relative expression of genes involved in the deposition of fibrotic scar tissue, under the TNF-α + IL-1β stimulation, an upregulation of TGF-β and FN1 mRNA levels was detected in fibroblasts cultured alone and after DPSC co-culture. FN1 is a common marker used to identify fibrocytes and fibroblasts [60]. Fibroblast differentiation is regulated by TGF-β, which has an autocrine and paracrine function. It drives fibroblast phenotypic switch into myofibroblasts by inducing α-SMA expression and collagen expression through the binding of smad3/smad4 to the TGF-β-responsive element in COL1A2 gene [61]. Interestingly, Western blot and confocal microscopy analysis revealed that at protein level, FN1 as well as α-SMA expression were downregulated in stimulated fibroblasts after DPSC co-culture. Furthermore, the significant increase of MMP-9 production by cytokine-stimulated co-cultured fibroblasts suggests a role in ECM remodeling promoted by DPSCs. As a matter of fact, MMP-9 is a matrix-degrading enzyme involved in several biological processes, including tissue remodeling and fibrosis resolution [62]. As TNF-α is a transcriptional activator for MMPs in dermal human fibroblasts, TNF-α higher levels observed in co-cultured cells likely contribute to the higher expression of MMP-9 that we observed.

In conclusion, our data suggest crosstalk between fibroblasts and DPSCs, where the latter are neither altered in their neural crest-related phenotype nor in their biological properties when exposed to inflammatory stimuli and indeed seem to modulate genes involved in inflammasome activation, ECM deposition, and, therefore, in wound healing and fibrosis. It is noteworthy that DPSCs exposed to distinct pro-inflammatory stimuli can enhance the transcription of genes involved in the inflammasome activation, whereas they can modulate a pro-fibrotic milieu reducing the ECM deposition and thus contributing to the recovery of tissue homeostasis. These results, even if preliminary, could pave the way for further studies to deepen the time-dependent interactions between fibroblasts and DPSCs and the role of DPSC in modulating fibrosis.

## Figures and Tables

**Figure 1 cells-13-00836-f001:**
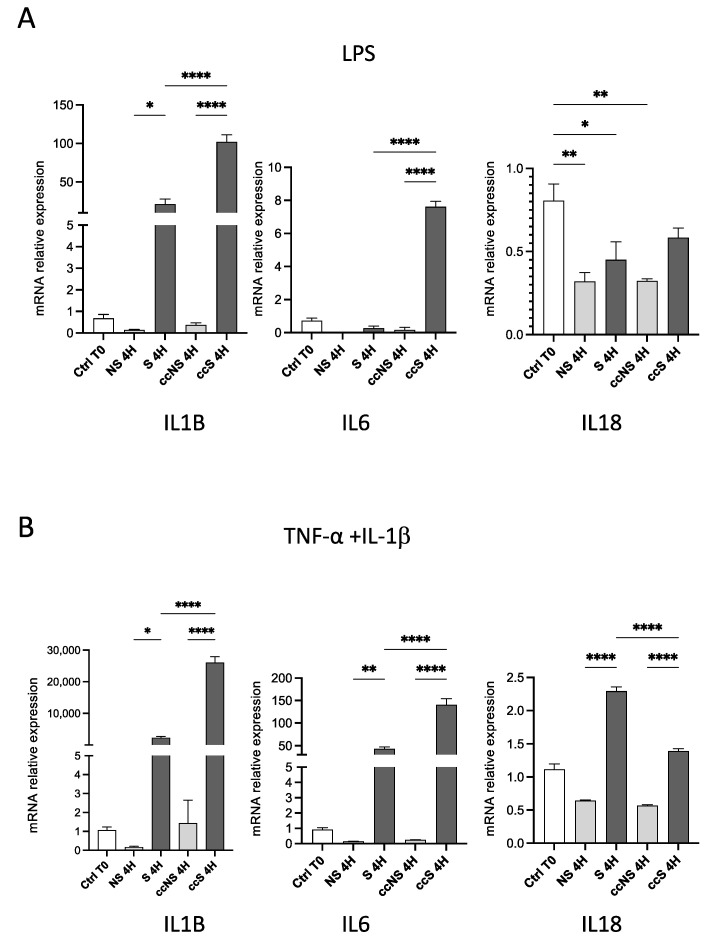
(**A**) Relative expression of the genes encoding pro-inflammatory cytokines IL-1β, IL-6, and IL-18 in fibroblasts treated with LPS (upper panels) or TNF-α and IL-1β (lower panels) for 4 h, alone or in co-culture with DPSCs. Data are reported as fold change respect to Ctrl T0, set to 1, and shown as mean ± SD of three independent experiments. (**B**) Relative expression of the genes encoding pro-inflammatory cytokines IL-1β, IL-6, and IL-18 in fibroblasts treated with TNF-α and IL-1β for 4 h, alone or in co-culture with DPSCs. Data are reported as fold change respect to Ctrl T0, set to 1, and shown as mean ± SD of three independent experiments. Ctrl: fibroblasts before stimulation, without co-culture; NS: non-stimulated; S: stimulated; ccNS: non-stimulated in co-culture with DPSCs; ccS: stimulated in co-culture with DPSCs. * = *p* < 0.05; ** = *p* < 0.01; **** = *p* < 0.0001.

**Figure 2 cells-13-00836-f002:**
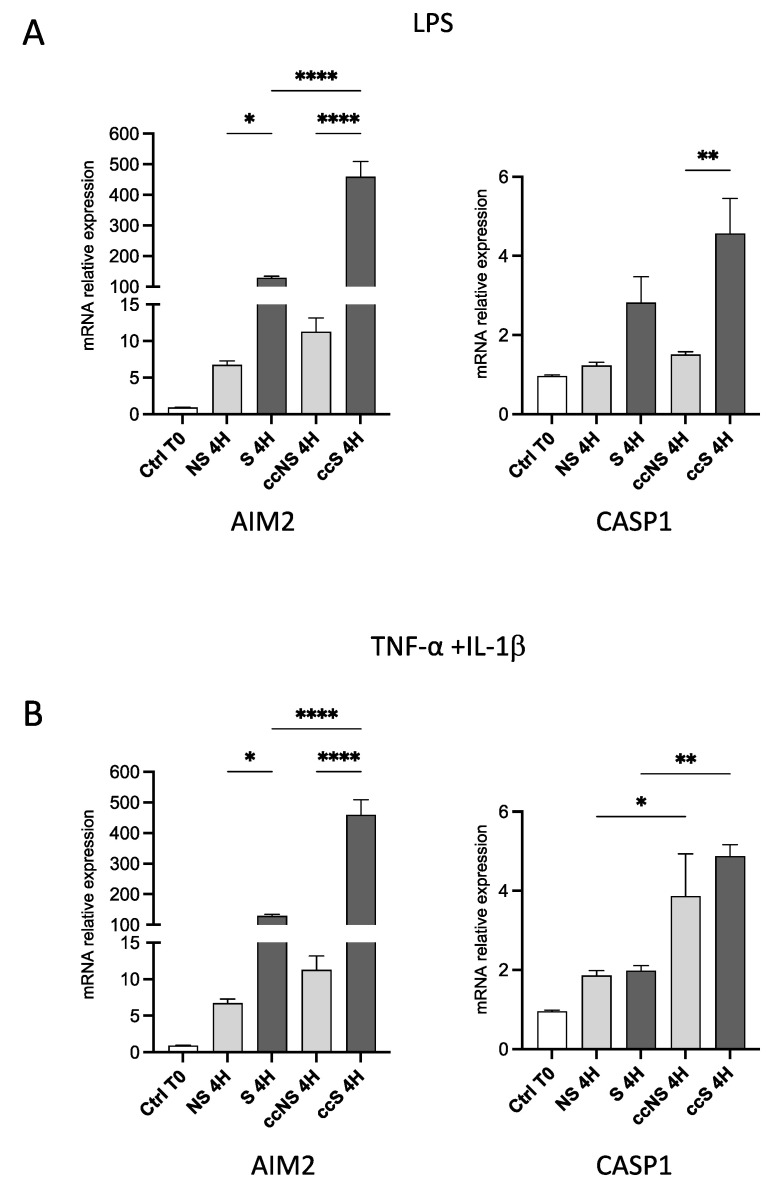
(**A**) Relative expression of *AIM2* and *CASP1* in fibroblasts treated with LPS for 4 h, alone or in co-culture with DPSCs. Data are reported as fold change respect to Ctrl T0, set to 1, and shown as mean ± SD of three independent experiments. (**B**) Relative expression of *AIM2* and *CASP1* in fibroblasts treated with TNF-α and IL-1β for 4 h, alone or in co-culture with DPSCs. Data are reported as fold change respect to Ctrl T0, set to 1, and shown as mean ± SD of three independent experiments. Ctrl: fibroblasts before stimulation, without co-culture; NS: non-stimulated; S: stimulated; ccNS: non-stimulated in co-culture with DPSCs; ccS: stimulated in co-culture with DPSCs. * = *p* < 0.05; ** = *p* < 0.01; **** = *p* < 0.0001.

**Figure 3 cells-13-00836-f003:**
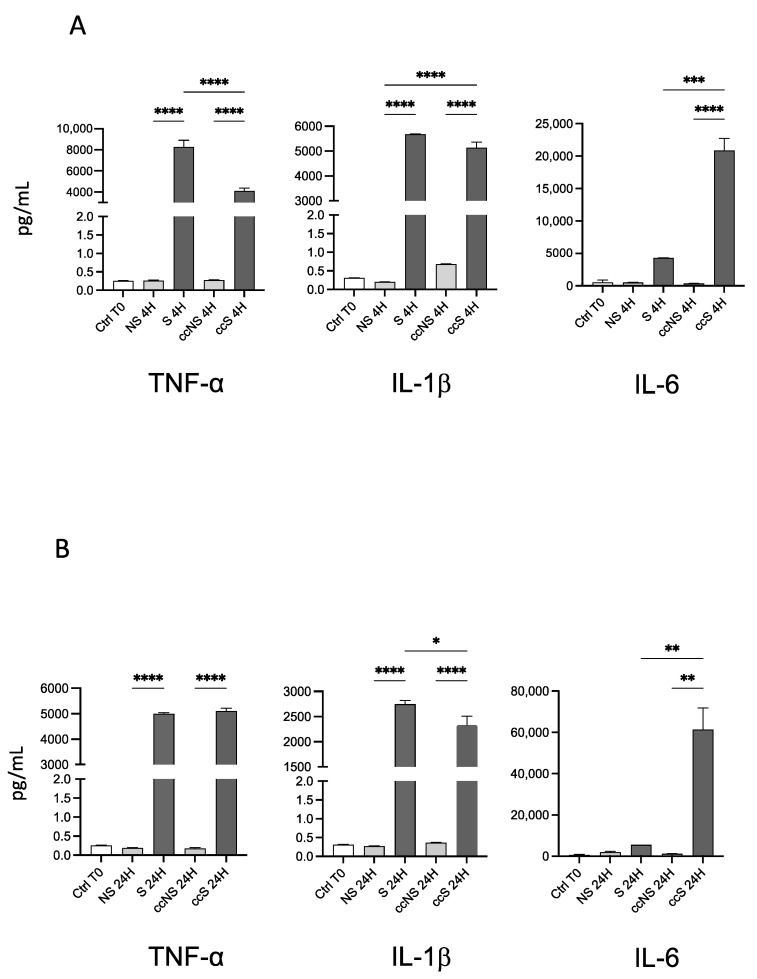
(**A**) Concentration of pro-inflammatory cytokines IL-1β, IL-6, and TNF-α in the supernatants of fibroblasts alone or in co-culture with DPSCs, after 4 h of treatment with TNF-α and IL-1β. Data are reported as fold change with respect to Ctrl T0, set to 1, and shown as mean ± SD of three independent experiments. (**B**) Concentration of pro-inflammatory cytokines IL-1 β, IL-6, and TNF-α in the supernatants of fibroblasts alone or in co-culture with DPSCs after 24 h of treatment with TNF-α and IL-1β. Data are reported as fold change respect to Ctrl T0, set to 1, and shown as mean ± SD of three independent experiments. Ctrl T0: fibroblasts before stimulation, without co-culture; NS: non-stimulated; S: stimulated; ccNS: non-stimulated in co-culture with DPSCs; ccS: stimulated in co-culture with DPSCs. * = *p* < 0.05; ** = *p* < 0.01; *** = *p* < 0.001; **** = *p* < 0.0001.

**Figure 4 cells-13-00836-f004:**
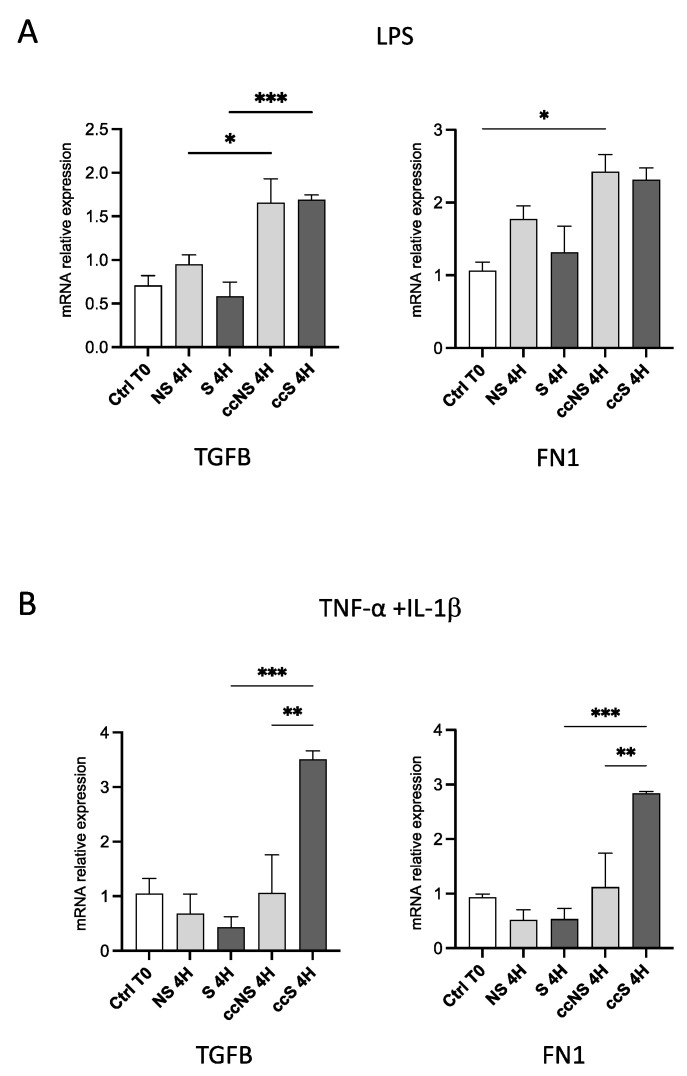
(**A**) Relative expression of the genes encoding TGF-β and FN1 in fibroblasts treated with LPS for 4 h, alone or in co-culture with DPSCs. Data are reported as fold change respect to Ctrl T0, set to 1, and shown as mean ± SD of three independent experiments. (**B**) Relative expression of the genes encoding TGF-β and FN1 in fibroblasts treated with TNF-α and IL-1β for 4 h, alone or in co-culture with DPSCs. Data are reported as fold change respect to Ctrl T0, set to 1, and shown as mean ± SD of three independent experiments. Ctrl T0: fibroblasts before stimulation, without co-culture; NS: non-stimulated; S: stimulated; ccNS: non-stimulated in co-culture with DPSCs; ccS: stimulated in co-culture with DPSCs. * = *p* < 0.05; ** = *p* < 0.01; *** = *p* < 0.001.

**Figure 5 cells-13-00836-f005:**
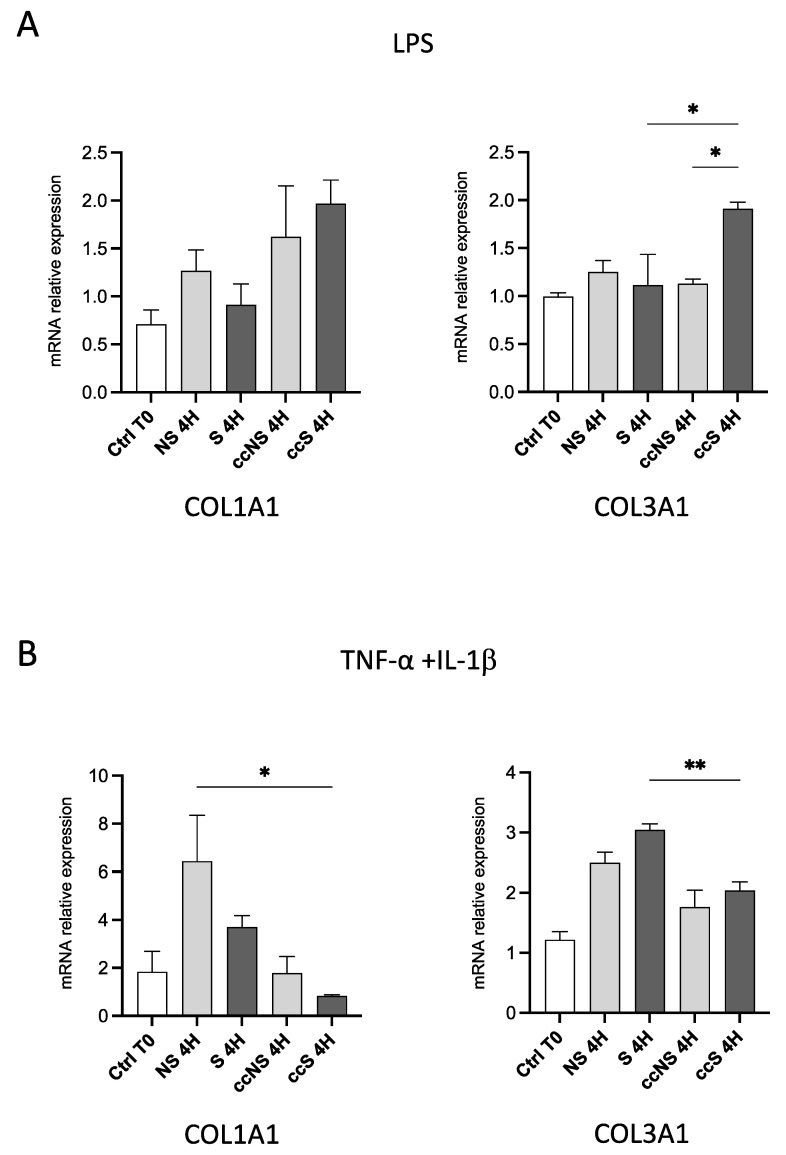
(**A**) Relative expression of the genes *COL1A1* and *COL3A1* in fibroblasts treated with LPS (upper panels) or TNF-α and IL-1β (lower panels) for 4 h, alone or in co-culture with DPSCs. Data are reported as fold change respect to Ctrl T0, set to 1, and shown as mean ± SD of three independent experiments. (**B**) Relative expression of the genes *COL1A1* and *COL3A1* in fibroblasts treated with TNF-α and IL-1β (lower panels) for 4 h, alone or in co-culture with DPSCs. Data are reported as mean ± SD of three independent experiments. Ctrl: fibroblasts before stimulation, without co-culture; NS: fibroblasts non-stimulated; S: fibroblasts stimulated; ccNS: fibroblasts non-stimulated in co-culture with DPSCs; ccS: fibroblasts stimulated in co-culture with DPSCs. * = *p* < 0.05; ** = *p* < 0.01.

**Figure 6 cells-13-00836-f006:**
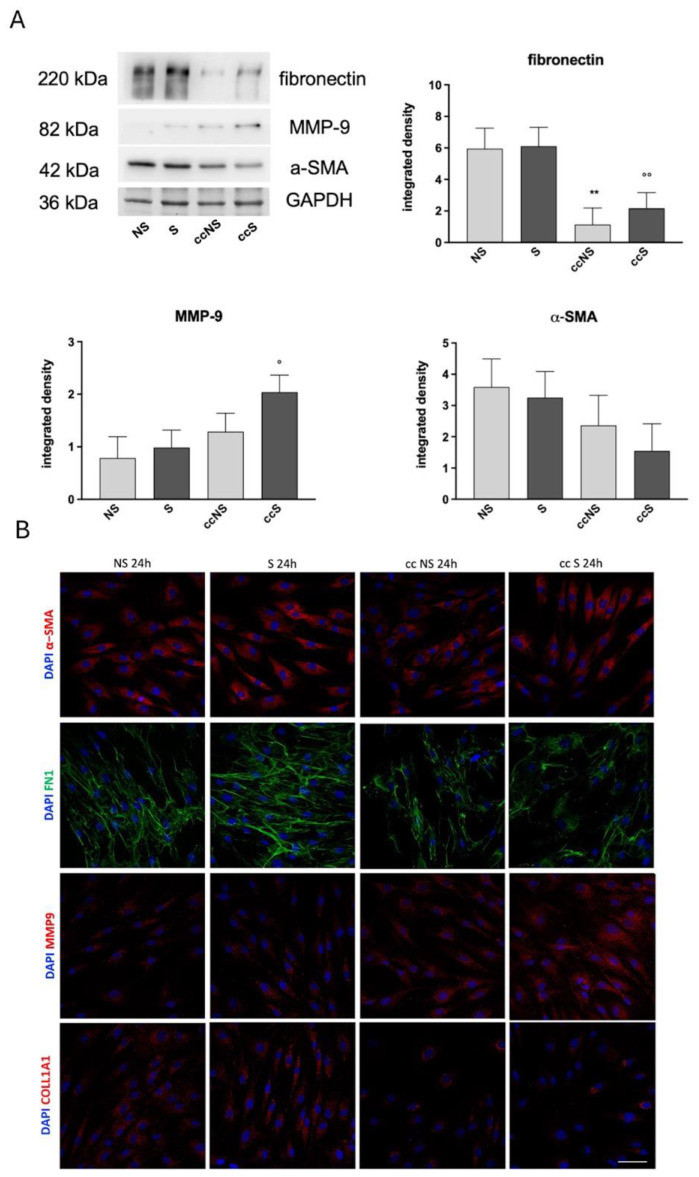
(**A**) Representative immunoblots and relative protein expression of Fibronectin (FN1), α-Smooth-Muscle-Actin (α-SMA) and Metalloproteinase 9 (MMP-9) in fibroblasts treated with TNF-α and IL-1β for 24 h, alone or in co-culture with DPSCs (**B**) Representative confocal microscopy images of fibroblasts treated with TNF-α and IL-1β for 24 h, alone or in co-culture with DPSC, labeled with anti FN1, anti-α-SMA, anti-COLL1A1 and anti-MMP-9 antibodies. Nuclei were counterstained with DAPI. NS: fibroblasts non-stimulated; S: fibroblasts stimulated; ccNS: fibroblasts non-stimulated in co-culture with DPSCs; ccS: fibroblasts stimulated in co-culture with DPSCs. One-way ANOVA was followed by Newman–Keuls post-hoc test, data are presented as mean ± SD of three independent experiments (n = 3). ** = *p* < 0.01 vs. NS; ° = *p* < 0.05, °° = *p* < 0.01 vs. S. Scale bar: 20 μm.

## Data Availability

The data sets analyzed during the current study are available from the corresponding author upon reasonable request.

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
