# Peer review of "Dental Pulp Stem Cells Modulate Inflammasome Pathway and Collagen Deposition of Dermal Fibroblasts"

_cells, 2024, doi:10.3390/cells13100836_

Round 1
Reviewer 1 Report
Comments and Suggestions for Authors
In this paper entitled "Dental Pulp Stem Cells Modulate Inflammasome Pathway and Collagen Deposition of Dermal Fibroblasts”, Giada Zanini et al. investigated DPSC and fibroblast interaction on inflammatory mediators expression and collagen protein.
The data are novel and somewhat interesting, however, this reviewer thinks that experimental designs and the conclusion may not be significant to the standard of Cells journal. It`s generally known that stem cells are secretory and affect the behavior of target cells in various ways. There are many studies of stem cell (and interacting cells) phenotype changes under inflammatory stimulation. The authors used a simple two cells culture system and examined the expression of common inflammatory molecules using the PCR and IHC without showing any mechanisms or in vivo study.
Stem cell behavior significantly alters depending on the concentration of inflammatory agents. The authors need to provide phenotype changes (or justifications) in different parameters.
Rather than just listing expressions of common inflammatory mediators, I would suggest pinpointing a major inflammatory molecule and seeking its mechanism.
Need higher-magnification IHC images (e.g. confocal) for Figure 6. Also, provide whether these increased expressions are from the fibroblasts or stem cells with a marker analysis (e.g. stem cell for STRO-1 & fibroblasts for FSP).
More importantly, for the mechanism, you may apply a key inflammatory agent inhibitor (or siRNA) without DPSC to see if this disrupts collagen protein expressions on fibroblasts.
Author Response
Please, see the attached file.

Reviewer 2 Report
Comments and Suggestions for Authors
This study is an interesting and valuable to the existing literature. The introduction, results and discussion are well written, however, some points need more explanation.
Comments
Material and Methods :
2.2. Cell Culture - “Fibroblasts and DPSCs were seeded on six-well plates at a density of 2,5x105 and 2x105, respectively”. Why the cells were seeded in different density?
- in co-culture the ratio between the different cells is important.
Results:
Figure 1 A legend is confusing for reader. Please correct because should be IL-1, IL-6 and IL-8 upper panels.
Figure 6 COLL1A1 staining panels; cc(NS 24h) and cc S 24h is poor quality.
I recommended this manuscript to minor revision for future process.
Round 2
Reviewer 1 Report
Comments and Suggestions for Authors
The authors addressed most concerns within limited time and resources, and the manuscript is acceptable to publish in its current form.
Author Response
We thank the reviewer for appreciating our effort to fulfill her/his requests, and for considering our manuscript acceptable for publication.